# Evaluation of the LightCycler^®^ PRO Instrument as a Platform for Rhesus D Typing

**DOI:** 10.3390/biomedicines12081785

**Published:** 2024-08-06

**Authors:** Helene Polin, Barbara Wenighofer, Nina Polonyi, Martin Danzer

**Affiliations:** Red Cross Transfusion Service of Upper Austria, Krankenhausstrasse 7, 4020 Linz, Austria

**Keywords:** blood group genotyping, transfusion medicine, Rhesus D antigen, *RHD* typing, LightCycler^®^ PRO

## Abstract

Rapid and reliable Rhesus D typing is crucial for blood donation centers. In instances of massive blood transfusion or reduced antigen expression, DNA-based phenotype prediction becomes mandatory. Our molecular *RHD* typing approach involves an initial real-time PCR for the most common aberrant *RHD* types in our region, *RHD*01W.1* (weak D type 1), *RHD*01W.2* (weak D type 2), *RHD*01W.3* (weak D type 3), and *RHD*07.01* (DVII). For comprehensive coverage, Sanger sequencing of *RHD* coding regions is performed in the case of PCR target-negative results. We evaluated the specificity and accuracy of these methods using the recently launched LightCycler^®^ PRO real-time platform. All findings demonstrated remarkable accuracy. Notably, the LightCycler^®^ PRO instrument offers a distinct advantage in data interpretation and integration via the HL7 interface. This study underlines the importance of including advanced molecular techniques in blood typing protocols, especially in scenarios where conventional serological methods may be insufficient.

## 1. Introduction

The Rhesus (Rh) blood group system is one of the most important blood group systems in clinical practice, primarily due to its role in blood transfusion and maternal–fetal medicine. Two different genes, *RHD* and *RHCE*, form the basis of this system. These genes are located on the short arm of Chromosome 1, with their 3′ ends facing in opposite directions [1]. 

The RhD protein forms the Rh complex with RhCE and the Rh-associated glycoprotein (RhAG), maintaining the integrity and flexibility of the red blood cell membrane. Structural models of the RhD protein predict 12 transmembrane helices passing through the lipid bilayer multiple times, contributing to its stability and function. Six extracellular loops are involved in the antigenic properties of the RhD protein [2].

The Rh blood group system includes the antigens D, encoded by the *RHD* gene, in addition to C, c, E, and e, encoded by the *RHCE* gene. Due to its high immunogenicity, D is known as the most clinically significant antigen among the Rh system. Individuals who lack the D antigen (RhD-negative) can develop antibodies against it if exposed to RhD-positive blood, leading to alloimmunization [2]. Besides RhD-positive and RhD-negative, there exists a large number of *RHD* alleles resulting in a variant D antigen phenotype [3,4]. In approximately 0.2 to 1 percent of routine samples, a weakened D antigen is determined. As standard procedure, blood donors with a serological weak D antigen are managed as RhD-positive, while patients and pregnant women are classified as RhD-negative [5]. Based on the knowledge that not all RhD types with serological weak D expression need to be treated as RhD-negative, further characterization and classification into one of the three categories of RhD variants—weak D, partial D or DEL—is recommended. 

Weak D types are defined by a quantitatively reduced expression of D antigen on the RBC surface, while the antigen remains qualitatively consistent. Most likely, the diminished antigen density is caused by folding disruptions during the integration of the RhD protein into the RBC membrane [6], which results from at least one amino acid substitution in the transmembraneous or intracellular part of the protein [7]. Weak D types 1, 2, and 3 are the most frequent variants in Europe. Patients with these types in a homozygous or hemizygous state can safely be considered D-positive without the risk of alloanti-D immunization [8,9].

In contrast, partial D variants harbor amino acid alterations on the extracellular protein segment. These changes affect single or several D epitopes or the 3-dimensional structure of that loop. Carriers of most partial D alleles can produce anti-D upon exposure to RhD-positive blood products [2]. Therefore, patients carrying partial D types should receive RhD-negative blood products. In Europe, weak partial D types 4.0 (*RHD*09.03.01*) and 4.1 (*RHD*09.04*) were described as frequent partial D alleles [2], but also DNB (*RHD*25*) [10], DVI type 1 (*RHD*06.01*) [11], and DVII (*RHD*07.01*) [12].

In blood donor management, DEL variants are of special interest. These types express only a very small amount of D antigen and are often mistyped as D-negative through standard serology. Due to the potential of anti-D immunization upon transfusion to D-negative recipients, carriers of DEL alleles should be removed from the D-negative donor pool [9,13]. While DEL variants are rare in Europeans, they are widespread in Asia [2].

For RhD status determination, molecular immunohematology is complementary or advantageous to serologic typing in the case of diminished antigen expression or after massive blood transfusion. Through *RHD* genotyping, allelic variants can be determined to allow for a safe and resource-saving transfusion strategy. Proper management of Rh compatibility helps prevent severe and potentially life-threatening complications, such as hemolytic transfusion reactions and HDFN [1,2].

The prevalence of aberrant RhD types varies among different ethnic groups [14,15]. In Caucasians, the *RHD* allele composition is dominated by only a few variant types. Weak D types 1, 2, and 3 contribute more than 65 percent of all serological weak D samples and approximately 90 percent of all molecular weak D samples in Central Europe [16,17,18,19,20]. 

Due to the comprehensive application of molecular characterization worldwide, the number of ongoing initial descriptions of novel *RHD* alleles [21,22,23] is considerable [3,4]. Given the significant amount of aberrant RhD types, it seems self-evident that commercial *RHD* typing kits are not capable of covering all associated genetic alterations. DNA sequencing may overcome these limits by providing detailed information about the entire coding sequence of the respective gene. This allows for the identification of all types of genetic variations, including single nucleotide polymorphisms (SNPs), insertions, deletions, and more complex structural variations. Therefore, Sanger sequencing, next-generation short-read sequencing, and Nanopore sequencing are advantageous for clarification of rare or novel RHD gene alterations [24]. 

We introduce a molecular-based *RHD* typing scheme for the diagnostics of transfusion recipients, blood donors, and pregnant women with reduced antigen density or serologic typing discrepancies. All PCR reactions are performed under usage of the recently launched LightCycler^®^ PRO real-time PCR instrument. This strategy is suited for the Upper Austrian region. Based on data derived from *RHD* Sanger sequencing during the last 20 years, we defined weak D types 1-3 as the most common weak D types and DVII as the predominant partial D variant. Therefore, the testing approach comprises an initial real-time PCR for *RHD*01W.1* (weak D type 1), *RHD*01W.2* (weak D type 2), *RHD*01W.3* (weak D type 3), and *RHD*07.01* (DVII) under identical conditions. Based on data collected earlier, we assume that more than 80 percent of all samples in need of complementary molecular work-up can be determined in the first step. According to the large number of assigned gene formations [3,4], further PCR testing of single nucleotide polymorphisms (SNPs) specific for RhD variants does not seem cost-effective in this setup. Therefore, we decided to clarify the molecular background of the remaining samples through Sanger sequencing of genomic DNA stretches representative of all 10 *RHD* exons.

The goal of this study was to assess the performance of our molecular *RHD* typing scheme using the LightCycler^®^ PRO real-time device and examine the Melting Curve Genotyping analysis as an option for SNP detection and characterization.

## 2. Materials and Methods

### 2.1. Setup

Orders for the multiwell plate setup were automatically generated from Excel-based amplification protocols and integrated into the LightCycler^®^ PRO Software v.1.0.2 as .CSV files via the HL7 (Health Level 7) interface. For this process, only the definitions of the well position, sample ID, assay name, run profile name, and sample role were required.

### 2.2. Sample Collection

A total of 24 samples derived from external quality assessment (EQA) schemes were included in this study. Each proficiency set of the INSTAND No. 235 Immunohematology A (molecular diagnostic) consisted of 4 samples [25]. All samples were sent between May 2019 and 2024 as purified leukocytic DNA and stored at −20 °C until their implementation in this study. The cohort included RhD-negative specimens (*RHD*01N.01*, n = 5), common RhD-positive samples (*RHD*01,* n = 8), and a total of 11 aberrant *RHD* alleles. Among these, weak D type 1 (*RHD*01W.1*, n = 2), weak D type 2 (*RHD*01W.2*, n = 1), weak D type 3 (*RHD*01W.3*, n = 1), and DVII (*RHD*07.01*, n = 1) were present, along with 6 different types of rare and very rare weak D, partial D, and DEL phenotypes.

The nucleic acid concentration was measured using an Eppendorf BioPhotometer D30 (Eppendorf AG, Hamburg, Germany) according to manufacturer’s instructions. For subsequent molecular analysis, each DNA sample was diluted to a concentration of 10 ng/µL in PCR-grade water. 

### 2.3. Real-Time PCR for Weak D Types 1, 2, and 3 and DVII

In-house real-time PCR was performed to confirm the presence of *RHD*01W.1*, *RHD*01W.2*, *RHD*01W.3,* and *RHD*07.01*. The method was developed according to specifications from the sequence-specific priming (SSP) procedure [26]. This technique relies on the exact matching of the 3′ end of one primer to a specific alteration, enabling polymerase chain reaction (PCR) amplification of a precise haplotype. Conversely, PCR amplification does not occur if the primers do not match the target sequence. To exclude false-negative results, an internal control fragment (ß-Globin) was co-amplified in each reaction. All assays were conducted under identical thermocycling conditions on a LightCycler^®^ PRO (Roche Diagnostics, Penzberg, Germany).

Each PCR reaction contained a 2 µL genomic DNA (gDNA) sample (20 ng), 5 µL (1x) of LightCycler^®^ 480 SYBR Green I Master (Roche Diagnostics, Mannheim, Germany), and primers as listed in Table 1 (TIB MOLBIOL, Berlin, Germany) in a total volume of 10 µL. PCR conditions were as follows: enzyme activation: 95 °C for 10 min; amplification 1 (10 cycles): 95 °C for 1 s (ramp: 4.4 °C/s), 65 °C for 5 s (ramp: 2.2 °C/s), 72 °C for 20 s (ramp: 4.4 °C/s); amplification 2 (25 cycles): 95 °C for 1 s (ramp: 4.4 °C/s), 61 °C for 5 s (ramp: 2.2 °C/s; single detection format), 72 °C for 20 s (ramp: 4.4 °C/s); melting: 95 °C for 1 s (ramp: 4.4 °C/s), 70 °C for 5 s (ramp: 2.2 °C/s), 95 °C (ramp: 0.11 °C/s; continuous detection format); cooling: 40 °C for 30 s (ramp: 2.2 °C/s). The detection format was set to ‘SYBR Green’, and the analysis protocol was ‘Melting Curve Genotyping’. This technique supports genotype classification through melt curve profiling. In standard mode, all PCR products were analyzed by comparing the samples’ melting curves to previously defined genotyping standards for *RHD*01W.1*, *RHD*01W.2*, *RHD*01W.3*, *RHD*07.01*, non-*RHD*01W.1* (target *RHD*01W.1* negative), non-*RHD*01W.2* (target *RHD*01W.2* negative), non-*RHD*01W.3* (target *RHD*01W.3* negative), and non-*RHD*07.01* (target *RHD*07.01* negative) by applying the Development Software v. 1.0.0. The defined genotypes could be edited manually and transferred into the laboratory information system via the HL7 interface. Furthermore, the Development Software supports the generation of a self-designed report containing essential information for data management, such as sample IDs, well positions, Tm values, and genotype results.

PCR-grade water was used as the no-template control (NTC) for every 96-well plate. The NTC was pipetted directly onto the amplification well without undergoing a previous extraction step. Negativity of the NTC ruled out nucleic acid contamination, therefore ensuring the reliability of the results. The NTC PCR reaction included all four primers used for internal controls (Table 1).

### 2.4. Sanger Sequencing of RHD Exons 1–10

Subsequently, *RHD* exons 1–10 and flanking introns were amplified using the LightCycler^®^ PRO instrument. Each 10 µL PCR reaction contained 5 µL (1x) of LightCycler^®^ 480 SYBR Green I Master, a 2 μL gDNA template (20 ng), and primers (TIB MOLBIOL) as shown in Table 2. To facilitate Sanger sequencing of PCR products, we added M13 forward (M13f-21) universal sequencing primer [29] to the 5′ end of each primer per PCR reaction (Table 2).

The PCR conditions were as follows: an initial denaturation step at 95 °C for 10 min; 10 cycles of denaturation at 95 °C for 15 s (ramp: 4.4 °C/s), annealing at 65 °C for 20 s (ramp: 2.2 °C/s), and extension at 72 °C for 70 s (ramp: 4.4 °C/s); followed by 25 cycles of denaturation at 95 °C for 15 s (ramp: 4.4 °C/s), annealing at 61 °C for 20 s (ramp: 2.2 °C/s), and extension at 72 °C for 70 s (ramp: 4.4 °C/s) with fluorescence detection in single mode; melting consisting of denaturation at 95 °C for 1 s (ramp: 4.4 °C/s), annealing at 70 °C for 5 s (ramp: 2.2 °C/s), and continuous temperature increase up to 95 °C (ramp: 0.11 °C/s; continuous detection format); followed by cooling: 40 °C for 30 s (ramp: 2.2 °C/s). The detection format was set to ‘SYBR Green’ and the analysis protocol was ‘Melting Curve Genotyping’. PCR products were analyzed through comparison to previously defined melting curves for each exon. 

Nucleotide sequences of PCR-positive templates were generated through cycle sequencing using the BigDye Terminator v1.1 Cycle Sequencing kit (Applied Biosystems, Carlsbad, CA, USA) and a Genetic Analyzer (SeqStudio; ThermoFisher Scientific, Waltham, MA, USA) following the manufacturer’s instructions. PCR products were initially treated with Exonuclease I (Thermo Fisher Scientific) and FastAP (Thermo Fisher Scientific) and purified using Sephadex G-50 Bio-Reagent (Merck, Darmstadt, Germany) before being sequenced using M13 forward (-21) universal sequencing primer (5′-CAGGAAACAGCTATGAC-3′). 

All electropherograms were visualized and edited using SeqScape Software v. 4.0 (Applied Biosystems). Before analysis, the quality of each raw DNA sequence was assessed through visual inspection of chromatograms. Quality values (QVs) greater than 20 indicate base calls with an error probability of 0.01 [30]. To ensure accurate and reliable sequencing data, only high-quality base calls (QV20 or higher) were validated automatically, whereas pure base calls with lower-quality values as well as mixed base calls were visualized and edited manually. 

Sequence reads were aligned to the *RHD* reference sequence NM_016124.6 [4]. For genotyping, all identified nucleotide and amino acid alterations were compared to those of listed alleles [3,4].

**Table 2 biomedicines-12-01785-t002:** Oligonucleotides for amplification of *RHD* coding regions.

PCR Reaction	Primer ID	Primer Sequence 5′-3′	Concentration [µM]	Tm [°C]
*RHD* Exon 1	RHD_E1_2f_M13	CAGGAAACAGCTATGACGCTTCCGTGTTAACTCCATAGAG	0.5	88.4
	RHD_E1_2r	GGGGGAATCTTTTTCCTT	0.5	
*RHD* Exon 2	RHD-E2_936_F6	ATGACAGTAACAGCACGCAC [31]	0.25	89.0
	RHD_i2+61R7_M13	CAGGAAACAGCTATGACTATCCCAGATCTTCTGGAACC	0.25	
*RHD* Exon 3	Ds3s_M13	CAGGAAACAGCTATGACGTCGTCCTGGCTCTCCCTCTCT [32]	0.25	86.5
	RH_E3_r1	GAGATGAGGATCTTGCTATGATG	0.25	
*RHD* Exon 4	RHD_E4_1f_M13	CAGGAAACAGCTATGACTATCAGGGCTTGCCCC	0.25	85.8
	RHD_E4_2r	TCAGACACCCAGGGGAAC	0.25	
*RHD* Exon 5	RH_E5_1f_M13	CAGGAAACAGCTATGACGACCTTTGGAGCAGGAGTG	0.15	88.7
	RHD_E5_2r	TGTGACCACCCAGCATTCTA	0.15	
*RHD* Exon 6	Ds6a_M13	CAGGAAACAGCTATGACCTTCAGCCAAAGCAGAGGAGG [32]	0.25	88.3
	Ds6-s	CAGGGTTGCCTTGTTCCCA [32]	0.25	
*RHD* Exon 7	RHD_E7_1f_M13	CAGGAAACAGCTATGACCCCCCTTTGGTGGCC	0.25	87.1
	RHD_E7_2r	CTTTGGTCTATACCTAGGTGGC	0.25	
*RHD* Exon 8	RHD_E8_3f	GGAGGCTCTGAGAGGTTGAG	0.25	87.2
	RH_E8_2r_M13	CAGGAAACAGCTATGACAATTATGTGATCCTCAGGGAAG	0.25	
*RHD* Exon 9	RHD_E9_2s_M13	CAGGAAACAGCTATGACTCCAGGAATGACAGGGCT	0.15	78.4
	RH_E9_2r	TTAAGTTCATGCACTCAAAATCTAT	0.15	
*RHD* Exon 10	RH_E10_1f_M13	CAGGAAACAGCTATGACAGAGATCAAGCCAAAATCAGTAT	0.15	85.6
	RHD_E10_1r	ATGGTGAGATTCTCCTCAAAG	0.15	

Underlined oligonucleotides indicate M13 primer tails.

## 3. Results

Plate orders were created through barcode scanning at the office workstation in Excel format and immediately transferred via the HL7 interface. After completing all pipetting work, the amplification was started, requiring no further manual input.

Through the use of PCR-SSP, specific nucleotides for *RHD*01W.2*, *RHD*01W.3,* and *RHD*07.01* were determined through real-time PCR for each sample, while *RHD*01W.1* was confirmed in two DNA eluates. These findings are fully consistent with the interpretation data provided by INSTAND, as detailed in Table 3. DNA amplification was successful across all samples, and false-negative results in PCRs negative for *RHD*01W.1*, *RHD*01W.2*, *RHD*01W.3,* and *RHD*07.01* were ruled out through proper amplification of the internal control fragment (Figure 1). 

According to the fundamental principles of PCR-SSP, the lack of detection for *RHD*01W.1-*, *RHD*01W.2-*, *RHD*01W.3-*, or *RHD*07.01*-specific results indicates the absence of these specific RHD genotypes. However, it does not provide any information regarding the presence or absence of the *RHD* gene itself.

For data analysis, the standard method of Melting Curve Genotyping was applied. All melting curve peaks generated during the process were compared to external melting standards derived from a previous run. Using these external standards, genotypes were accurately assigned without requiring any corrections. The results were compiled as shown in Figure 2 and then transmitted via the HL7 protocol into our database system for record-keeping.

In a total of five samples, no *RHD*-specific templates could be generated through PCR prior to DNA sequencing. These samples were suspected to harbor an *RHD* gene deletion, specifically the *RHD*01N.01* allele. Additionally, in another sample, genomic DNA could not be amplified for *RHD* exons 6 to 9, raising suspicion of the presence of the *RHD*04.03* allele, which correlates with a DIV type 3 phenotype. Further verification through PCR targeting *RHD* exons 1 to 10 with primers located in exonic regions is necessary to confirm the presence of these alleles.

The remaining 18 samples with successful amplification of *RHD* exons 1 to 10 underwent DNA sequencing (Table 3). Evaluation of these 180 chromatograms showed that base calls for all reactions were above QV 20. Genotype identification through alignment of the generated sequences to a reference allele confirmed the previous real-time PCR results for *RHD*01W.1*, *RHD*01W.2*, *RHD*01W.3,* and *RHD*07.01*, showing full concordance.

In six samples, no alterations were identified compared to the reference sequence, indicating the presence of the *RHD*01* allele in a homozygous or hemizygous state. Additionally, the alleles *RHD*01W.5* (weak D type 5) and *RHD*11* (weak partial type 11) were each identified in one sample. Among the cohort, we detected one sample with an associated DEL phenotype, defined as *RHD*01EL.01*. Furthermore, two partial D alleles were detected, *RHD*19* (DHMi) and *RHD*25* (DNB). Although the result published by INSTAND indicated *RHD*25*, we obtained this allele in a heterozygous state. Precisely, the associated SNP G>A at position 1063 was determined to be in trans to an unaltered c.1063G (Table 3). 

## 4. Discussion

One of the major challenges in clinical transfusion practice is the prevention of anti-D alloimmunization in recipients of blood products. Simultaneously, D-negative erythrocyte concentrates must not be wasted and should be reserved for patients who require them [1,2]. Additionally, both the applicability and cost-effectiveness of the methodology used are crucial considerations for blood banks and hospitals [33,34].

Serologic phenotyping remains the gold standard for Rhesus D testing due to its low costs and fast turnaround. However, a safe transfusion strategy cannot be assigned in cases of diminished or ambiguous D antigen expression. Molecular analysis is widely adopted to overcome these limitations by defining the allelic background [1,2,5,7]. Through determination of the genotype, an associated phenotype can be predicted, thus supporting transfusion strategies and the management of anti-D immunoglobulin [2,5].

A comprehensive workflow for molecular *RHD* typing was developed using the newly introduced LightCycler^®^ PRO real-time device. To enhance the speed of typing, the initial testing step involves PCR-SSP to detect the most common aberrant RhD types prevalent in our region, specifically weak D types 1, 2, and 3 and DVII. DNA samples negative for these specific SNPs are subsequently characterized through sequence-based typing (SBT) of *RHD* coding regions in a second step.

To evaluate the effectiveness of our new typing approach, we included samples from previously conducted proficiency programs spanning a 6-year period. As external quality assessment schemes set the minimum standard for diagnostic tests [25], they seem valuable for ensuring the quality of this innovative technique.

The results derived using real-time PCR assays targeting SNPs correlated to *RHD*01W.1*, *RHD*01W.2*, *RHD*01W.3,* and *RHD*07.01* showed complete concordance with those obtained from proficiency panel assessments. It is essential to clarify that this real-time PCR assay was specifically developed to detect prominent *RHD* SNPs in samples exhibiting reduced D antigen expression. Consequently, a negative result for the targeted allelic variants does not necessarily imply the absence of the *RHD* gene. To ensure accurate determination of *RHD* negativity, especially in pre-transfusion scenarios, it is advisable to supplement this assay with exon-specific reactions.

For comprehensive evaluation, each sample underwent DNA sequencing, regardless of the PCR outcome. This approach contrasts with the routine typing protocol, where only a subset of samples would typically be sequenced. In a standard testing scenario, samples testing positive for *RHD*01W.1*, *RHD*01W.2*, *RHD*01W.3,* or *RHD*07.01* through real-time PCR would not undergo further characterization.

DNA sequencing results were fully concordant with published data as well as those generated through initial PCR, affirming the accuracy and reliability of our new typing approach. Moreover, although INSTAND as the proficiency panel distributor reported a *RHD*25* (DNB) genotype associated with a single nucleotide alteration c.1063G>A in one sample, we identified this alteration in a heterozygous state. This finding highlights the capability of DNA sequencing to characterize the entire coding sequence, thus identifying additional variations throughout the *RHD* gene.

To facilitate sequencing, each amplification primer for *RHD* exons 1 to 10 was tailed with an M13 sequence. Application of a universal primer is a common practice in molecular biology, enabling sequencing of multiple PCR templates like *RHD* exons 1 to 10 with only 1 primer instead of 10 or even 20 primers in cases of sequencing in both directions. This approach reduces handling errors and lowers costs by optimizing workflow efficiency. Despite the inherent risk of primer dimer formation associated with this technique [35], we successfully obtained high-quality sequences with well-resolved electropherograms.

The use of SYBR Green I Master proved effective for specific amplification and discrimination of different targets. Intercalating dyes facilitate the specific differentiation of PCR products through melting curve analysis tools like Melting Curve Genotyping or Tm Calling. SYBR Green I exhibits a cycle-dependent increase in fluorescence in the presence of double-stranded DNA (dsDNA) without sequence specificity. In contrast to TaqMan Probes, which discriminate PCR products during amplification, intercalating dyes require a separate melting curve analysis step after amplification [36]. Although this additional analysis consumes more time compared to TaqMan approaches, it avoids the need for cost-intensive probe addition. The melting behavior of a PCR template is influenced by its length and sequence but predominantly by its GC content. Unlike gel electrophoresis, where products of the same length but with different GC/AT ratios may not be distinguished, melting curve analysis can differentiate these variations [36]. This feature facilitates both the development and composition of PCR assays. Additionally, using an intercalating dye like SYBR Green I simplifies the process by eliminating the need to switch between channels for the target and the internal control, thereby reducing potential errors.

In this study, we utilized a melting curve analysis tool to detect and distinguish PCR fragments of four different SNPs corresponding to RhD variants and ten distinct *RHD* exons. Melting Curve Genotyping demonstrated robust performance in terms of detection and discrimination of different targets. This evaluation method was deemed user-friendly and less prone to error by associating predefined melting curve patterns with specific *RHD* variants. In addition, this technique eliminates the need for manual interpretation of melting temperatures (Tms) as required in Tm Calling or the visualization of bands in gel electrophoresis. When handling tubes or plates containing amplified products for agarose gel electrophoresis, significant amounts of aerosolized DNA can disperse into the laboratory environment, potentially leading to contamination issues if spread over reagents [37]. Approximately 80 percent of our samples are typed as *RHD*01W.1*, *RHD*01W.2*, *RHD*01W.3,* or *RHD*07.01* through initial PCR and do not demand further characterization. In this large proportion, our approach mitigates the risk of amplicon contamination by eliminating the need for further handling of amplified DNA. Therefore, resources are conserved and hands-on time is reduced.

Notably, the LightCycler^®^ PRO multiwell plate instrument offers a distinct advantage in throughput compared to the capillary-based LightCycler 2.0 device with the ability to amplify only 32 samples per run. It should be taken into account that the LightCycler^®^ PRO specifically requires the use of barcoded plates, which are available exclusively in white. Non-transparent plates can introduce errors due to their lack of visibility, making the pipetting process more prone to inaccuracies. These issues can be effectively mitigated by automating the pipetting process. Automation not only improves the accuracy and reliability of the results but also enhances the efficiency of laboratory operations, making it essential for molecular diagnostics [38].

In this study, the LightCycler^®^ PRO demonstrated high correctness and precision in genetic analysis. The import of pre-filled plate setups proved highly suitable for routine processing, significantly simplifying and shortening the startup process. Additionally, the integration of collected genotypes into our in-house database was facilitated by Melting Curve Genotyping analysis and the HL7 connection, ensuring accurate data transfer without the risk of human error. This capability not only enhances efficiency but also supports compliance with legal requirements for data retention.

Furthermore, the ability to generate individualized reports strengthens our ability to maintain comprehensive records in accordance with regulatory standards. This feature ensures that all genetic testing results are securely documented and readily accessible as needed.

It is noteworthy that all PCR reactions were performed in a total volume of only 10 µL each. Using smaller volumes reduces the amount of reagents required, thereby lowering overall costs. Moreover, when dealing with limited or precious samples, smaller volumes allow for more reactions to be performed with the same amount of sample material [39].

This method was designed to serve as a foundation for laboratories specializing in the molecular biological clarification of serologically suspicious RhD samples. The PCR-SSP method can and should be extended or modified based on the regional distribution of predominant RhD types. According to published studies, the implementation of SNPs related to DNB and the inclusion of DVI types instead of DVII appear to be beneficial enhancements for some regions [10,11].

In summary, we present a novel real-time-based typing approach for RhD variants with questionable D antigen status. The detection of *RHD*01W.1*, *RHD*01W.2*, *RHD*01W.3,* and *RHD*07.01* parallel to an internal control fragment is performed using PCR-SSP. This real-time PCR supports detection of up to 80 percent of variant D samples in less than one hour from DNA extraction. Sanger sequencing of *RHD* coding regions completes the testing in cases where PCR results are target-negative, enabling the detection of all assigned *RHD* alterations and even characterizing novel alleles without limitations.

A major advantage of the presented technique is its accuracy. The LightCycler^®^ PRO instrument supports semi-automated genotyping and integration of data and therefore supports safe and reliable RhD routine diagnostics.

## Figures and Tables

**Figure 1 biomedicines-12-01785-f001:**
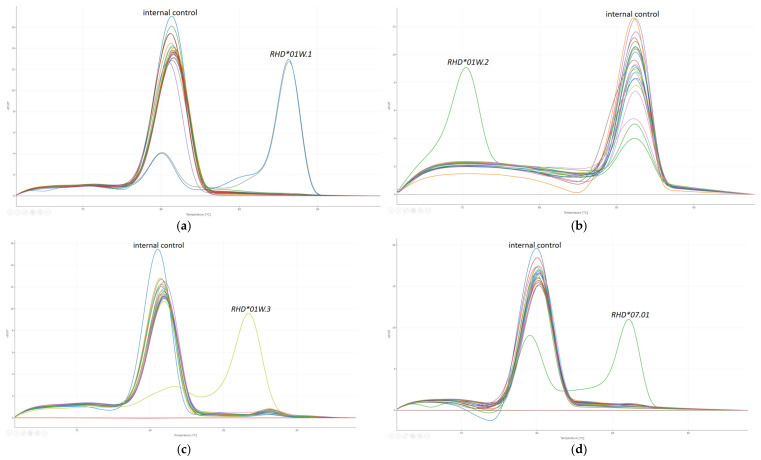
Specific melting curves of (**a**) *RHD*01W.1*, (**b**) *RHD*01W.2*, (**c**) *RHD*01W.3*, and (**d**) *RHD*07.01* PCR reactions. Different color lines represent distinct samples.

**Figure 2 biomedicines-12-01785-f002:**
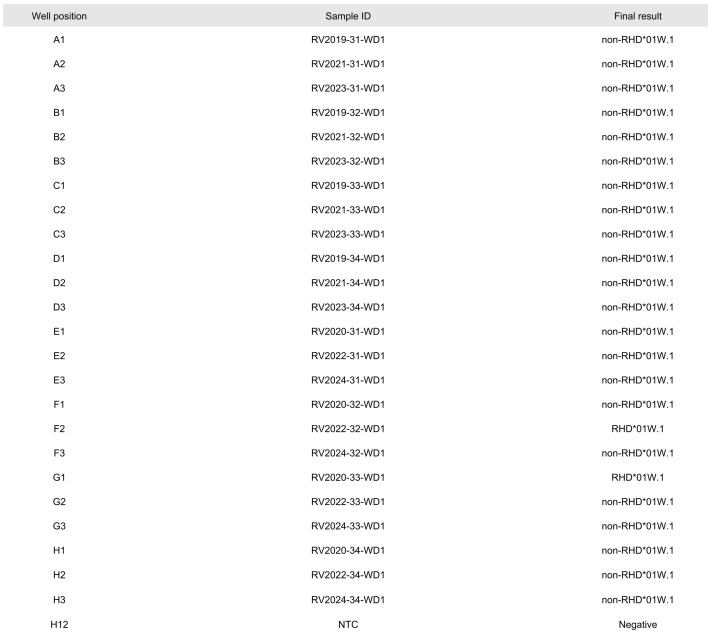
Melting Curve Genotyping analysis report of *RHD*01W.1*-specific PCR reaction.

**Table 1 biomedicines-12-01785-t001:** Oligonucleotides introduced in the real-time PCR specific for *RHD*01W.1*, *RHD*01W.02*, *RHD*01W.03,* and *RHD*07.1*. Small characters of primer sequences indicate intron positions, while capital letters indicate exon positions of the nucleotides.

Specificity	rs No	Primer ID	Primer Sequence 5′-3′	Concentration [µM]	Tm [°C]
*RHD*01W.1*	rs121912763	WD1-809G-f	acacgctatttctttgcagACTTATGG [20]	0.25	83
(NM_016124.6:c.809T>G)		WD1-r	GGTACTTGGCTCCCCCGAC [20]	0.5	
*RHD*01W.2*	rs71652374	WD2-1154C-f	ctccaaatcttttaacattaaattatgcatttaaacagC [20]	0.5	74.4
(NM_016124.6:c.1154G>C)		WD2-r	gtgaaaaatcttacCTTCCAGAAAACTTGGTCATC [20]	0.5	
*RHD*01W.3*	rs144969459	WD3-8G-f	acagagacggacacaggATGAGATG [20]	0.25	86.4
(NM_016124.6:c.8C>G)		WD3-r	CTTGATAGGATGCCACGAGCCC [20]	0.5	
*RHD*07.01*	rs121912762	RHD-E2-201-f	GCTTGGGCTTCCTCACCTCG [27]	0.25	85.8
(NM_016124.6:c.329T>C)		RHD-329C-r	ccaccatcccaatacCTGAACG [28]	0.15	
ß-Globin		G107F	CTGGGCAGGTTGGTATCA [13]	0.25	80.7
(chr11:5234243-5234349) ^1^		G107R	GAGAGTCAGTGCCTATCAGAAAC [13]	0.25	
ß-Globin		globin F	CAACTTCATCCACGTTCACC [13]	0.25	86
(chr11:5226949-5227216) ^2^		globin R	GAAGAGCCAAGGACAGGTAC [13]	0.25	

^1^ Introduced in PCRs for *RHD*01W.01*, *RHD*01W.03,* and *RHD*07.01*. ^2^ Co-amplified with *RHD*01W.2*.

**Table 3 biomedicines-12-01785-t003:** Results released from INSTAND and derived from this study.

Year	Sample	INSTAND-Published Results	This Study		
			*SSP* PCR Results	Sanger Sequencing Results	Determined Genotype (Phenotype)
2019	31	*RHD* gene (normal)	negative for target SNPs	NM_016124.6:c.1_1254=	*RHD*01* (normal D antigen)
	32	*RHD* gene (normal)	negative for target SNPs	NM_016124.6:c.1_1254=	*RHD*01* (normal D antigen)
	33	DIV	negative for target SNPs	no amplification of *RHD* exons 6–9	*RHD*04.03 ** (DIV type 3)
	34	weak D type 5	negative for target SNPs	NM_016124.6:c.446C>A	*RHD*01W.5* (weak D type 5)
2020	31	*RHD* gene absent on both chr	negative for target SNPs	no amplification of *RHD* exons 1–10	*RHD*01N.01* * (D−)
	32	*RHD* gene absent on both chr	negative for target SNPs	no amplification of *RHD* exons 1–10	*RHD*01N.01* * (D−)
	33	weak D type 1	*RHD*01W.1*	NM_016124.6:c.809T>G	*RHD*01W.1* (weak D type 1)
	34	DVII	*RHD*07.01*	NM_016124.6:c.329T>C	*RHD*07.01 (DVII)*
2021	31	*RHD* gene absent on both chr	negative for target SNPs	no amplification of *RHD* exons 1–10	*RHD*01N.01* * (D−)
	32	*RHD(K409K)*	negative for target SNPs	NM_016124.6:c.1227G>A	*RHD*01EL.01 (Del)*
	33	*RHD* gene (normal)	negative for target SNPs	NM_016124.6:c.1_1254=	*RHD*01* (normal D antigen)
	34	*RHD* gene (normal)	negative for target SNPs	NM_016124.6:c.1_1254=	*RHD*01* (normal D antigen)
2022	31	DHMi	negative for target SNPs	NM_016124.6:c.848C>T	*RHD*19* (DHMi)
	32	weak D type 1	*RHD*01W.1*	NM_016124.6:c.809T>G	*RHD*01W.1* (weak D type 1)
	33	*RHD* gene absent on both chr	negative for target SNPs	no amplification of *RHD* exons 1–10	*RHD*01N.01* * (D−)
	34	*RHD* gene (normal)	negative for target SNPs	NM_016124.6:c.1_1254=	*RHD*01* (normal D antigen)
2023	31	*RHD* gene (normal)	negative for target SNPs	NM_016124.6:c.1_1254=	*RHD*01* (normal D antigen)
	32	*RHD* gene absent on both chr	negative for target SNPs	no amplification of *RHD* exons 1–10	*RHD*01N.01* * (D−)
	33	DNB	negative for target SNPs	NM016124.6:c.[1063G>A];[1063=]	*RHD*25/RHD*01* (DNB het.)
	34	weak D type 2	*RHD*01W.2*	NM_016124.6:c.1154G>C	*RHD*01W.2* (weak D type 2)
2024	31	*RHD* gene (normal)	negative for target SNPs	NM_016124.6:c.1_1254=	*RHD*01* (normal D antigen)
	32	*RHD* gene (normal)	negative for target SNPs	NM_016124.6:c.1_1254=	*RHD*01* (normal D antigen)
	33	weak D type 3	RHD*01W.3	NM_016124.6:c.8C>G	*RHD*01W.3* (weak D type 3)
	34	weak D type 11	negative for target SNPs	NM_016124.6:c.885G>T	*RHD*11* (weak partial type 11)

* Confirmation of deletion by exon-specific primers required.

## Data Availability

All of the analyzed data are included in the article. The raw data can be obtained upon request.

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
