# Peer review of "Evaluation of the LightCycler® PRO Instrument as a Platform for Rhesus D Typing"

_biomedicines, 2024, doi:10.3390/biomedicines12081785_

Round 1
Reviewer 1 Report
Comments and Suggestions for Authors
In the manuscript entitled "Evaluation of the LightCycler® PRO instrument as a platform for RHD typing" the authors besides presenting the advantage of LightCycler® PRO in genetic analyses, they also justified the importance of including advanced molecular techniques in blood typing protocols, especially in scenarios where conventional serological methods may be insufficient.
Even though, the methodology and results are sound and clear my only concern is the small number of samples used to evaluate the presented pipeline. However, the combination of different methodologies in detecting false positive or negative results seems adequate and mitigates or even eliminates the possibility of a sample with an ambiguous D expression to escape undetected.
.
Author Response
Comments: In the manuscript entitled "Evaluation of the LightCycler® PRO instrument as a platform for RHD typing" the authors besides presenting the advantage of LightCycler® PRO in genetic analyses, they also justified the importance of including advanced molecular techniques in blood typing protocols, especially in scenarios where conventional serological methods may be insufficient. Even though, the methodology and results are sound and clear my only concern is the small number of samples used to evaluate the presented pipeline. However, the combination of different methodologies in detecting false positive or negative results seems adequate and mitigates or even eliminates the possibility of a sample with an ambiguous D expression to escape undetected.
Response: Thank you for your review and your positive assessment. We fully understand your concerns regarding the low number of samples. Unfortunately, we were unable to include the validation data of the patient and donor samples in the publication, as there was not enough time before the deadline for the special issue to submit the application to the ethics committee. We therefore had to limit ourselves to the publication of the EQA samples.
Reviewer 2 Report
Comments and Suggestions for Authors
I read with attention and interest the work: Evaluation of the LightCycler PRO instrument as a platform for RHD typing by Polin H and colleagues.
Personally I believe that the need to combine phenotyping with genotyping is a topic of great importance in immunohematology and that this need is particularly relevant in the study of RHD.
The work seemed well written to me even if in some parts it appears to be rather verbose.
In the introduction section the lines could be eliminated:
28-29, 37-39, 46-48, 56,57, 70,71. furthermore, the paragraph between line 80 and line 89 should be rewritten as it appears to be difficult to understand.
The Materials and Methods section appears to be rather verbose but it constitutes the core of the study and therefore I do not think it is appropriate to make cuts in the text that could make it difficult to understand.
In the discussion section on line 275 the term that tested should be replaced with resulted.
The paragraph from line 332 to line 340 should be deleted and the paragraph from line 341 to line 352 should be rewritten and shortened.
Comments on the Quality of English Languageminor revision required
Author Response
Many thanks for your review and your positive assessment. We tried to implement your correction suggestions as good as possible with regard to the lower limit of 4000 words.
In the introduction section the lines could be eliminated:
28-29 was deleted as suggested
37-39 we respectfully disagree to delete this sentence, as we think that it is important for the understanding of the Rhesus system in the context
46-48 was deleted as suggested
56,57 was deleted as suggested
70,71 was deleted as suggested
furthermore, the paragraph between line 80 and line 89 should be rewritten as it appears to be difficult to understand.
The paragraph was re-written
In the discussion section on line 275 the term that tested should be replaced with resulted.
“that tested” was replaced with “resulted”, as requested.
The paragraph from line 332 to line 340 should be deleted and the paragraph from line 341 to line 352 should be rewritten and shortened.
We respectfully disagree to delete the paragraph about the contamination problematic with gel electrophoresis. We think it is important to overthink and reflect the disadvantages of standard PCR-SSP. We think the reader might improve from the presented knowledge.
The paragraph from line 341 to 352 was rewritten and shortened, as requested.